# Laboratory Tests Using Distributed Fiber Optical Sensors for Strain Monitoring

**DOI:** 10.3390/s25020324

**Published:** 2025-01-08

**Authors:** Rodrigo Moraes da Silveira, Marcelo Buras, André Luiz Delmondes Pereira Filho, Juliana Ferreira Fernandes, Marcos Massao Futai

**Affiliations:** 1Civil Construction Department, Polytechnic Center, Federal University of Parana, Curitiba 80530-000, Brazil; 2Civil Engineering Department, Polytechnic School, Pontifical Catholic University of Parana, Curitiba 80215-901, Brazil; 3Institute of Technology for Development, Curitiba 80215-090, Brazil; marcelo.buras@lactec.com.br; 4Civil Engineering Department, Polytechnic School, University of Sao Paulo, São Paulo 05508-220, Brazil; delmondes@usp.br (A.L.D.P.F.); julianaffernandes@alimini.usp.br (J.F.F.); futai@usp.br (M.M.F.)

**Keywords:** distributed fiber optics, tensile tests, single-mode fiber

## Abstract

Using fiber optics as a tool for different kinds of geotechnical monitoring can be highly attractive and cost-effective when compared to conventional instruments, such as piezometers and inclinometers, among others. A single fiber optic cable may cover a larger monitoring area compared to conventional instrumentation and allows for monitoring more than one physical quantity with the same fiber optic cable. The literature provides several different examples of distributed fiber optic systems usage. For using any sensor, a calibration curve and parameters are required. In the case of strain sensors, calibration is required to derive strain values from the frequency measurement quantity. However, fiber optic sensor cable manufacturers do not often provide cable calibration parameters, and researchers should consult the specialized literature. This article thus presents a bench adjusted for tests with single-mode fiber optic cables, as well as results of tensile tests for defining the function of strain variations in two different optical fiber cables manufactured by different companies using two different distributed interrogators. This paper also proposes a methodology for calibrating fiber optic cable deformation. A few manufacturers of fiber optic cables aim at civil engineering applications. Therefore, we propose a calibration methodology to show the possibility of obtaining calibration parameters of any fiber optic cable, even those manufactured for telecommunications purposes and not only for cables manufactured for civil engineering use. Thus, researchers will not be restricted to the acquisition of special cables for their applications. The results allowed us to conclude that the application of calibrated fiber optic sensors to experimental pile foundations permits the evaluation of the load–displacement behavior of these elements under different loading conditions.

## 1. Introduction

The fiber optic industry underwent significant development starting in the late 1980s. Besides communication, this technology is also used for measuring and monitoring deformation, displacement, acceleration, pressure, temperature, and chemical properties, among others [1]. Distributed fiber optic monitoring systems have become significantly popular in geotechnical applications as they can provide distributed strain and temperature measurements with high accuracy along any monitored area without any gaps.

Technological evolution has provided increasingly reliable, robust, and economical solutions. Geotechnical instrumentation emerged and began to be applied between the 1930s and 1940s [2]. In the first 50 years following the emergence of instrumentation, major changes were reported in this field. The predominance of the use of simple, mechanical, and hydraulic instruments gave way to safer and more complex instruments with electrical and pneumatic transducers. In 1988, Dunnicliff warned of the advent of automatic data acquisition systems and computerized data processing.

Geotechnical instrumentation provides data that helps decision-making at any stage of a project to reduce or avoid catastrophic situations. Soils and rocks have anisotropic and heterogeneous characteristics that, added to the complexity of local hydrological conditions, lead to a series of uncertainties regarding soil deformation [3]. The latter is one of the great challenges of geotechnical engineering regarding monitoring the deformation of materials at the surface and subsurface levels. Several conventional monitoring technologies are used, but these only allow for knowing measures taken occasionally, with a limited amount of data. With the aid of fiber optic technology, different monitoring methods can be proposed that even allow for establishing a soil displacement profile with greater precision and safety.

The application of fiber optic sensors to geotechnical monitoring has been used worldwide. A fiber optic cable can cover the monitoring of a much larger area when compared to conventional instrumentation. Works by [4,5,6,7,8,9] are some examples that prove the potential of the technology.

Over the last three decades, several technologies and optical properties have been approached for evaluating different magnitudes, measuring variations in the properties of light transported in optical fibers, such as its intensity, frequency, wavelength, phase, or polarity [10,11].

From a metrological point of view, optical fibers’ high stability, precision, and resolution stand out. They are also long-lasting and made of a non-corrosive and chemically stable material. For applications in geotechnical engineering, they have total immunity to electromagnetic interference and the total absence of electrical noise, besides allowing signal transmission over long distances without any degradation or interference.

Punctual or local optical sensors allow for measuring physical quantities at discrete places or points. They correspond to sensors with a measurement base of a few millimeters. In the case of structural monitoring, they are particularly suitable for assessing local effects, capturing the concentration of effects with high accuracy. As an example, the Bragg sensors and the Fabry–Perot interferometers can be mentioned. Distributed optical sensors are a unique aspect of fiber optic sensors. This configuration allows for evaluating a given physical quantity along the entire length of the sensor continually in space. Variation profiles can be achieved over developments that can reach several kilometers. They are particularly suitable sensors for detecting and locating damage in highly developed structural elements, such as foundations, dams, pipelines, and geotechnical works in general. Brillouin microscopy and Raman sensors have boosted the creation of distributed fiber optic sensors.

Among the technological advantages of fiber optic sensors, the most notable are (i) sensing several signals over a single optical fiber (multiplexing data), (ii) ease of reading the signals (good value signal versus noise), (iii) measurements over long distances (remote sensing), (iv) immunity to electromagnetic fields, (v) absence of spark, and (vi) low weight and low material reactivity.

A single fiber optic cable may cover a larger monitoring area, compared to conventional instrumentation, and monitor more than one physical quantity with the same fiber optic cable. This makes using fiber optics highly attractive and cost-effective.

Consulting the literature, it is possible to find several different examples of distributed fiber optic systems used for monitoring the performance of a soil-nailed slope [12,13], tunnels [14,15], mass movements [16,17], slope monitoring [18], and geotechnical structural elements [19] such as pile foundations [20,21,22,23,24,25,26] and landfills [27], in addition to those aforementioned. For this, the interrogator, the optical sensor cable, and the installation technique must be properly adjusted to guarantee the quality of the measurement results.

Analogously to conventional sensors, a sensor calibration curve/parameter is required to derive strain values from the frequency measurement quantity. However, a sensor calibration curve/parameter often is not provided by manufacturers [28].

The Institute of Engineering Geodesy and Measurement Systems, IGMS, developed a unique calibration facility [28], enabling highly precise, fully automatic calibration of strain sensors with lengths of up to 30 m under stable laboratory conditions. Prior to field measurements, individual calibrations of several samples of the sensing cables used are usually carried out to reliably determine the frequency-to-strain relation. Results of various calibrations of different sensing cables for geotechnical applications from Solifos AG (e.g., types V3 and V9) are shown in [29].

Aiming at field applications for strain monitoring piles, calibrations of a Brazilian and a UK fiber optical sensor cable were carried out to determine the frequency and temperature-to-strain relation. For field application, the optical fiber will be installed with a pre-tensile, considering that piles are usually under compression. For monitoring pile strain, a perfect adhesion of the cables to the piles must exist. This is not the issue of this study, and it will be part of a future research study.

This article covers all the steps undertaken to define the function of strain variations in two different single-mode optical fiber cables using two different distributed interrogators. To fulfill this objective, this paper shows the calibrations that were carried out in Lactec (city of Curitiba, Parana State, Brazil), results, and conclusions. The authors of [28] presented a work focused on the calibration results of the fiber optic-based monitoring system; however, the article in question did not observe detailed details of the methods used in the calibration tests of strain and temperature performed, aiming at possible reproduction. Therefore, this article proposes a methodology for calibrating the deformation of fiber optic cables since the calibration parameters are provided by the cable manufacturers, which can result in errors depending on the application.

It should be noted that there are few manufacturers of fiber optic cables aimed at applications in civil engineering. Therefore, the calibration methodology proposed in this article is intended to transmit to the academic community the possibility of obtaining calibration parameters of any fiber optic cable, even those manufactured for telecommunications purposes and not only for cables manufactured with an intended use in civil engineering. Given this, researchers will not be restricted to the acquisition of special cables for their applications.

## 2. Principle of Fiber Optic Strain Sensing

As commented by [30], the measuring principle of distributed fiber optic strain sensing is based on the fact that after sending a light pulse by a powerful light source (laser) into a glass fiber, a very small proportion of this light is backscattered at each point along the fiber. The scattered light undergoes a frequency shift, called the Brillouin frequency shift, which depends on the strain and temperature variations. Figure 1 illustrates the scattered light spectrum. There are other types of measuring principles, but in this article, only what was used to carry out the laboratory tests is discussed.

The analyzed section of the fiber optic is determined with the commercially available Brillouin Optical Time Domain Reflectometer (BOTDR) system or Brillouin Optical Time Domain Analysis (BOTDA) system. With the knowledge of the light speed in the material, it is possible to identify the corresponding Brillouin spectrum of each section. The reading unit (interrogator) used in the laboratory tests, called DTSS (Distributed Temperature and Strain Sensing) and DSTSs (Distributed Strain and Temperature Sensors), records both Stokes and anti-Stokes light portions of the Brillouin spectrum, shift, and power. Analysis of these data allows for strain and temperature evaluation along the fiber optic cable.

## 3. Material and Methods

For cable fiber optical calibration, the materials and methods used are presented below.

### 3.1. Optical Cables and Interrogator Unit

The DTSS and the DSTS interrogators used in calibrations tests are a monitoring system developed by Sensornet (Watford, Hertfordshire, UK) and OZ Optics (Ottawa, ON, Canada), which measures the Brillouin spectrum with a resolution of 1.02 m along the fiber. With the analysis of the data obtained, it is possible to obtain strain data and temperature at all points along the cable [31].

In the optical cable calibration tests, two optical fiber distributed interrogators were used, including a DTSS interrogator manufactured by Sensornet and a DSTS interrogator manufactured by Oz Optics. The two interrogators record Brillouin frequencies for later calculations to convert the frequency into temperature and/or strain. In other words, the DTSS interrogator has BOTDR technology that uses Brillouin spontaneous backscattering. This backscatter allows for measuring deformations and/or temperature at any point along the optical fiber, with a spatial resolution of 1.02 m. In BOTDR technology, a single laser beam is pulsed at one end of the fiber. A small percentage of this backscattered light returns to the emitting source by the spontaneous backscattering of the Brillouin spectrum. Spectrum Brillouin frequency analysis is a consequence of changes in strain and/or temperature, while the travel time of the backscattered light pulse determines the measurement position along the length of the fiber. Figure 2A shows the Sensornet DTSS interrogator.

The DSTS interrogator has both technologies in its system, BOTDR and BOTDA. BOTDA technology utilizes Brillouin-Stimulated Backscatter. The main difference is that two laser beams are injected on opposite sides at the ends of the fiber. A continuous wave beam is injected at one end, while a laser pulse is injected at the other end of the fiber. In BOTDA, energy transfer occurs from one beam to the other, but the loss in the continuous wave beam is what is used for data interpretation. Although BOTDA technology requires access to both ends of the fiber, it has some advantages, such as a spatial resolution of up to 0.04 m, reading intervals with shorter acquisition time, and the possibility of monitoring up to 100 km of fiber. Figure 2B shows a photograph of the OZ Optics DSTS interrogator. As already mentioned, to use optical fiber or fiber optic cables in experiments and subsequent use in the experimental field, it is necessary to determine the parameters of calibrated strain measurements.

The DSTS interrogator (Oz Optics) uses the BOTDA method (Brillouin Optical Time Domain Analysis), and the DTSS interrogator (Sensornet) uses the BOTDR method (Optical Time Domain Reflectometer).

The methods differ in terms of how the optical fiber is connected to the interrogator. The first has a spatial resolution of less than 1 m, 0.1 °C for temperature measurements, and 2 με for strain measurements, in which the optical fiber is connected at both ends, causing the reading to be performed in the loop way, starting at one end and ending at the other. In the BOTDR, the fiber is connected only at one end; it has a spatial resolution of 1 m, 0.8 °C for temperature measurements, and 16 με for strain measurements, according to the manufacturer’s catalog.

The objective of using two interrogators simultaneously in the calibration methodology presented here was to seek knowledge about obtaining the same calibration parameters regardless of the interrogator used. In addition, it is known that with the BOTDA method, data with better spatial resolution can be obtained in relation to the BOTDR. Table 1 presents a summary of the characteristics of the cited interrogators.

The first optical cable used in the tests to determine the cable calibration parameter for strain measurements is sold by Sensornet (Figure 3A). This fiber optic cable is 6.5 mm in diameter, and it is made up of 4 optical fibers, two of which are multimode (green and brown) and two are single-mode (blue and orange). It has tensile reinforcement elements such as kevlar fibers surrounded by an external protection layer, as can be seen in Figure 3B. The maximum tensile strength of this fiber optic cable, informed by the contractor, is 3 kN, and the maximum deformation is 20,000 µε (equivalent to 2%).

The second optical cable used in the tests to determine the cable calibration parameter for strain measurements is sold by the manufacturer Furukawa (Curitiba, Brazil). The fiber optic cable is made up of tight buffer, single-mode optical fibers. The optical fibers have a primary coating of acrylate and a secondary coating of thermoplastic material. On the set of fibers, tensile elements of dielectric wires are placed. The set of fibers is protected against water penetration and has an outer layer of thermoplastic material that does not propagate flame and is weather-resistant. It has a nominal external diameter of 5.6 mm and consists of 6 optical fibers. The maximum tensile strength of this fiber optic cable, informed by the contracting party, is 1.85 kN. In Figure 3B, there is an illustration of the cable.

### 3.2. Conventional Sensors

For displacement measurements, two inductive linear displacement transducers (LVDT—Linear Variable Differential Transformer) were used; one was WA200 (with a measurement capacity of 200 mm), and the other one was WA300 (with a measurement capacity of 300 mm), both supplied by HBM (Hottinger Baldwin Messtechnik, Sao Paulo, Brazil). The maximum linearity deviation is 0.2% in relation to the sensitivity, which is 80 mV/V (Figure 4A).

Tensile force readings on the sample were taken by an HBM load cell S9M, with a capacity of 5 kN, nominal sensitivity of 2 mV/V, and accuracy class 0.02, which provides the values to the control systems (Figure 4B).

The test temperature control was carried out by three air conditioners with a frequency inverter and was performed by the average of the temperature read by seven T-type thermocouples (Figure 4C) installed along the sample, approximately 1650 mm from each other. For tests at temperatures above 26 °C, a heating system consisting of fourteen electrical resistances was used. The control of air conditioners and resistances was performed via test software (LabVIEW NXG 5.1 by National Instruments, Austin, TX, USA). All conventional sensors were previously calibrated in metrology laboratories according to specific technical standards.

### 3.3. Cable Anchoring System

An optical cable anchoring system was developed for the cable tensile test. It is important to point out that this development was carried out exclusively based on experimental tests tentatively carried out until obtaining what is presented here. When the loss of the optical signal was perceived with the cable pressed, the anchoring system was reassessed until reaching the dimensions shown below. There is no knowledge of a similar development since it can be said that it is a very personal development according to the dimensions of the fiber optic cable, which have different characteristics according to the purpose of the application. It is known that fiber optic cable manufacturers have cable calibration systems; however, these procedures are still confidential since attempts were made to know them, which was not possible.

The system used consists of two rectangular metal plates, measuring 23 cm by 12 cm on each side and 1 cm thick. The plate has a recess with roughness for the positioning of the cable, with a diameter slightly smaller than the diameter of the cable. This system is installed at both ends of the cable on the test bench. Figure 5A–D show photographs of this system.

For the positioning of the LVDT sensors, a support plate was developed that allows for adjusting the height of the sensors in relation to the cable. On this plate, there is an LVDT fixation system and a trolley to support its rod. This trolley is built on wheels so that friction when moving the cable is minimized.

To support the optical cable, a U-shaped metallic profile was developed with screws along its length that allow the height of the profile to be adjusted in relation to the height of the optical cable, thus preventing the cable from being deformed by its own weight. Figure 6 illustrates a section of this system.

### 3.4. Bench Details

The calibration bench used is 14 m long, and it can perform tests with tensile of up to 200 kN and with a total stroke of up to 1500 mm in length. The bench’s tensile equipment is equipped with a tensile control system, whose physical components are a servomotor and a reducer.

Signals from all force, displacement, and temperature transducers are stored on an NI SCXI 1000 datalogger and HBM’s MGA II, Sao Paulo, Brazil. The microcomputer receives the measurement readings through a PCI-6289 card, from National Instruments (NI), connecting an internal PCI bus to a microcomputer. This board also performs the function of sending temperature control signals to the air conditioners and tensile signals to the servomotor.

With the automation of the entire process, operator interference during tests is minimized, contributing to the uniformity of procedures. All measured data and control parameters are acquired and analyzed in real time with a graphic presentation. The test software was developed in LabView graphical language. Figure 7 shows the schematic drawing of the workbench.

### 3.5. Test Bench Preparation

The sensor monitoring data (LVDTs, load cell, and thermocouples) were collected in the same time interval as the optical interrogators through a routine developed in LabView. Special care was taken to activate the equipment together and achieve synchronization. The times of the three acquisition systems (test bench, DTSS, and DSTS) were synchronized. The strain increments were programmed for a strain variation of 15,000 με with a time interval of 9 min for each loading. These values were defined according to the maximum tension supported by the anchorage system identified in experimental tests.

The loads were performed by the person responsible for the test, while the sensor recordings were performed automatically. The temperature control (air conditioners) of the room where the tests were performed was also automatically controlled by the developed program in LabView.

To start the test, the cable was completely relaxed on the platform of the test bench, and then a manual adjustment of the engine displacement was performed until the strain variation was perceptible in the DTSS optical interrogator. At this moment, the deformation value in the cable was noted, which was around 1700 με, in the section of the cable under test. Figure 8 shows the initial condition of the cable, with the cable without strain applied.

After the mentioned adjustments, the test was started with strain in steps of 1500 με until reaching the value of 15,000 με. These strains were controlled based on the displacements measured by the LVDTs. For each of the three programmed test temperatures, 20 °C, 30 °C, and 40 °C, three repetitions were performed. In the third repetition of each temperature, the test bench was programmed to reach a value of 20,000 με, the deformation limit value read by the optical interrogators.

### 3.6. Test Procedure

The procedure for carrying out the tests is described below. Figure 9 shows a sketch of the test layout for the optical cable calibration, indicating the minimum length of each section. It can also be seen in this sketch that a single cable is connected to two optical interrogators, simultaneously, in the DTSS (BOTDR) and in the DSTS (BOTDA). It is important to mention that the DSTS and DSTS nomenclature are used according to the nomenclature defined by the different manufacturer companies (Oz Optics and Sensornet).

This connection is possible due to the two single-mode optical fibers of the cable used in the tests (blue and orange fibers). This test configuration is not common; however, the purpose of using two interrogators was to verify if the calibration parameters obtained using the different equipment used simultaneously in the tests were similar.

It should be noted that the minimum lengths indicated in Figure 9 were based on the DTSS interrogator acquisition system (BOTDR), where the spatial resolution is approximately 1 m. For smaller spatial resolutions (BOTDA system), the minimum lengths may be smaller and must be evaluated on a case-by-case basis.

For the optical cable calibration tests, a cable length of approximately 60 m was required. For the BOTDR system, the optical cable was connected to the optical interrogator; then, a cable section must pass through a temperature reference section. In this reference section, the deformation and temperature of the cable must be known. Thus, a section of loose cable (without deformation) was submerged in water at a known temperature (ambient temperature) so that during the tests, the temperature variation in this section was minimal. On the test bench, approximately 13 m of cable were installed. For each section of the cable (interrogator to the reference; reference to the test bench; and test bench to the end of the cable), it is recommended to leave a length greater than 5 m so as not to interfere with the reading section. For the BOTDA system, the cable still needs to return to the interrogator, so approximately 20 m more cable was needed.

After positioning the cable on the test bench, the anchoring systems, displacement transducers, and temperature sensors were installed on the cable. A load cell was installed on the bench for measuring tensile forces during calibration tests.

To install the anchoring systems on the cable, six screws from the system were tightened little by little and alternately. During the tightening of the screws, the torque was measured with the aid of a torquemeter. In previous tests, it was verified that the minimum torque applied so that the cable does not slip during tensile and the optical loss is minimized should be 4.5 N.m. Figure 10 shows the torquemeter used to install the anchoring system on the cable. It should be noted that the torque value indicated in the figure in question is only illustrative; it is not the value used to tighten the screws for installing the anchoring system.

The displacement sensors (LVDT) were installed at approximately 11.6 m from each other and 70 cm from the cable anchorage. Therefore, to calculate the strain, the distance between the displacement sensors was used; the distance between the cable anchorages was not used (Figure 11).

Seven temperature sensors (thermocouples) were installed on the cable, spaced every 1.80 m from each other. The sensors were fixed with the aid of self-fusion adhesive tape. These sensors helped control the activation of three air conditioners used to maintain a constant temperature in the room where the tests were performed through an interface developed in LabView. Figure 12 shows two thermocouples installed in the optical cable for monitoring temperature control during the tests. The load cell was installed between the test bench tensile system and the optical cable anchorage, as can be seen in Figure 13.

The temperature control of the test bench, which was carried out with the aid of three air conditioners and an electrical resistance, equivalent in length to the length of the cable, was installed inside the bench. The temperature was monitored by the average of the 7 thermocouples installed along the length of the cable. The increase in temperature to meet the range of 20 °C to 50 °C, provided for in the tests, was carried out in steps of 3 °C every 12 min. In the calculations, the temperature of the cable in the final 3 min of each step was discarded, as in this time interval, the addition to the next test temperature step was carried out.

Thus, the stabilization of the room ambient temperature at 20 °C was carried out with the aid of air conditioners. This stabilization was scheduled a day before each test. Up to the temperature of 26 °C, only air conditioners were used for heating. For the other temperature levels, the electrical resistance of the bench was used until the test limit temperature of approximately 50 °C was reached. It should be noted that for both the air conditioners and for the electrical resistance, the temperature control was carried out by the computer program developed in LabView.

The optical interrogators were configured to record strain measurements simultaneously over a 3 min time period. This time was based on the interrogators’ minimum acquisition time. For DTSS, the minimum acquisition rate defaults to 2 min, and the spatial resolutions are 1.0202 m. For DSTS, the acquisition rate depends on equipment configuration parameters, such as light pulse length (10 ns) and spatial resolution (0.32 m), number of scans to obtain a measurement (10,000), and frequency increment between successive sweeps (5 MHz). With the values indicated in parentheses, the minimum acquisition rate was 3 min. Therefore, both interrogators were programmed to take a reading every 3 min.

## 4. Data Analysis

This section shows the analysis and discussion of the results of nine tests carried out to determine the optical calibration parameter of the optical cables. The tests were carried out at three different ambient temperatures, 20 °C, 30 °C, and 40 °C, which are expected temperatures for operating conditions.

For each temperature, the tests were repeated three times; thus, nine tests were performed in total, as previously mentioned. The tensile force was monitored in the tests only to control the maximum tensile force supported by the cable, i.e., 3 kN. In this article, the results of this monitoring are not presented.

The determination of the optical cable calibration parameter for deformation measurements was carried out by linear regression, where the variations in the frequency of the Spectrum of Brillouin are correlated with the variations in the deformations imposed on the optical cable, read from displacement sensors (LVDTs).

Prior to the calibration tests, an analysis of the optical loss caused by crushing the cable in the anchoring systems was carried out. The optical loss was graphically analyzed based on the response of the OTDR (Optical Time Domain Reflectometer) signal, where it was possible to identify and quantify the signal loss in an optical cable.

Based on experience using the equipment, it was verified that the OTDR signal level, in order to have a good result, must be greater than 0.25 a.u. (astronomical unit), where the value 1 a.u. indicates that 100% of the signal is being transmitted, that is, there are no optical losses. The value corresponding to 0 a.u. indicates that no optical signal is being transmitted, that is, total optical loss, which can be caused by a critical angle or identification of the end of the optical cable.

Figure 14 shows the conceptual interpretation of the OTDR response signal, where the vertical axis represents the OTDR value and the horizontal axis represents the cable length. The slope of this plot is the natural attenuation (optical loss) along the length of the cable.

OTDR analyses were performed only with the Sensornet DTSS interrogator; in the DSTS equipment from OZ Optics, it is not possible to perform this type of analysis. Inside the DTSS interrogator, there is a section (stretch). This section comes from the factory. It is approximately 415 m. Between sections 305 m and 360 m, the OTDR signal has 100% of the signal (1 a.u.); in the 360 m length, there is an optical loss resulting from splicing of the optical fiber carried out along the installation, where the signal decreased to 0.93 a.u. up to length 415 m. From length 415 m, the OTDR signal is from the fiber that is connected to the equipment. It should be noted that the splice inside the equipment is performed due to the hardware connections of the equipment itself; it is not a fiber that was broken and spliced for repair. Figure 15 shows a graph based on the comments made in this paragraph.

The graphs presented below show the Brillouin frequency on the vertical axis and the strain on the horizontal axis. The angular coefficient of the line obtained by linear regression is the optical calibration parameter obtained for the deformation measurements.

The following items present and discuss the results obtained in the tests carried out for the three programmed ambient temperatures of 20 °C, 30 °C, and 40 °C. For the calibration tests of the optical cable at different temperatures, the first step consisted of stabilizing the temperature of the laboratory. The average time for the temperature to stabilize at 20 °C was approximately 1 h, for 30 °C, 1.5 h, and 40 °C, 2 h.

After the temperature stabilized, the test started with the verification of the response of the OTDR signal. The graph in Figure 16 shows an example of the result obtained from the OTDR response at the beginning of the test for the three tests carried out at a temperature of 20 °C. Evidently, these analyses were performed for all tests.

Based on the graph shown in Figure 16, it is possible to observe that there was an optical loss at the two cable anchorage points (lengths from 300 m to approximately 420 m). The cited optical losses ranged from 0.93 a.u. to 1.00 a.u. and did not make the tests unfeasible, as they have values greater than 0.25 a.u (ideal OTDR value for the tested cable length). It is also noted that the optical loss is greater in the first test and decreases in the second and third tests. This reduction in optical loss is probably due to the rearrangement of the fibers inside the cable after its tensile since the same stretch of cable is used in the tests.

After verifying the response of the OTDR signal, the test was then started, applying controlled deformations for the frequency measurement. Three tests were carried out at the same temperature in order to obtain the cable calibration parameter for strain measurements using the DTSS interrogator and the DSTS interrogator. The parameters were obtained through linear regression of the average deformations controlled by LVDTs and the variation in the frequencies obtained.

### Tensile Tests

This section presents the results of the tensile tests performed on the bench presented in Section 3.4. Tests were carried out on the fiber optic cable produced by Furukawa (Brazilian cable) and by Sensornet (UK cable). The tests consisted of varying the temperature and strain to be possible to compare the parameters obtained (angular coefficient).

Figure 17 and Figure 18 show the results obtained from carrying out deformation tests for different ambient temperatures (20 °C, 30 °C, and 40 °C) with the cable produced by the company Furukawa. The tests were performed with the DTSS and DSTS interrogators. The results are presented in terms of deformation versus monitored frequency.

It is important to emphasize that for the analysis of the results, it is not necessary to impose an offset. Changing the data with the line passing through the origin will evidently change the calibration coefficient. The readings are interpreted from the variation in the frequency of the Brillouin spectrum; thus, the coefficient “b” of the equation y = ax + b is annulled. The value used is just the “a” coefficient. The tests were conducted at different temperatures to determine if they would influence the calibration parameters.

Based on the calibration parameters obtained in the tests (Figure 17 and Figure 18), Table 2 presents a comparison of the results. Using the calibration equations obtained, the Brillouin spectrum frequency was varied between 0 and 1000 MHz for each test condition (20 °C, 30 °C, and 40 °C) and for each interrogator (DTSS and DSTS). In the table, “DTSS Difference” is the sub-tensile between the largest and smallest deformation obtained for the tests with the DTSS interrogator; “DSTS Difference” is the sub-tensile between the largest and smallest deformation obtained for the tests with the DTSS interrogator; and “DTSS and DSTS Difference” is the sub-tensile between the largest and smallest strain obtained, considering the values obtained by both the DTSS and DSTS interrogators. The highest and the lowest values in the table are highlighted in red and blue, respectively.

Analyzing the results presented in Table 2, the highest variations in deformations are observed to occur for the highest values of the Brillouin frequency shift; the highest deformations differences occur for the DSTS interrogator. Therefore, the average parameter obtained for the DTSS interrogator is 0.0502 MHz/με; for the DSTS interrogator, it is 0.0506 MHz/με, and the average parameter between the interrogators is 0.0504 MHz/με. These variations may have occurred due to assay settings, even though the readings on the two interrogators were simultaneous. In addition, for field applications, the differences in deformations monitored by the different equipment are negligible.

For each temperature, the same stretch of cable was always used, and the analyses did not consider whether there was permanent/residual deformation due to cable tension. Another factor that may have influenced the results of the tests is the initial state of deformation of the cable. During the tests, the initial position of the press was not carefully controlled; we only checked if the cable was loose at the beginning of the tests.

Also, the cable in the tests was suspended without a support, which certainly caused an initial deformation due to its own weight. The lack of a support may also have interfered with the strain readings, measured in the LVDTs sensors, since there is a device attached to the cable for carrying out the tests. This can explain the differences between the test results and the Sensornet calibration informed. Finally, the deformation of the Brazilian cable observed using each interrogator was the same at temperatures of 30 °C and 40 °C, which means the temperature at that level does not change the deformation.

Therefore, this explains why, in Table 2, the values of columns 3, 4, and 6, 7 are the same. It also explains why the third, fourth, sixth and seventh columns are red and blue.

Figure 19 and Figure 20 show the angular coefficients from the tests carried out on the Sensornet cable for each of the temperature conditions (20 °C, 30 °C, and 40 °C) with the DTSS and DSTS interrogators. These angular coefficients are the calibration parameters of the tested optical cable to obtain deformation data (LVDT) from the frequency of the Brillouin spectrum.

Based on the calibration parameters obtained in the tests (Figure 19 and Figure 20), Table 3 was prepared, like Table 2.

Table 3 shows that the highest variations in deformations occurred for the highest values of the frequency of the Brillouin spectrum and that the highest differences in deformations occurred for the DSTS interrogator. Therefore, the average parameter obtained for the DTSS interrogator is 0.0506 MHz/με; for the DSTS interrogator, it is 0.0511 MHz/με, and the average parameter between the interrogators is 0.0509 MHz/με.

As in the previous analysis, the variations may have resulted from the test configurations. For each temperature, the same stretch of cable was always used, and the analyses did not consider whether there was permanent/residual deformation due to cable tension. Finally, the cable construction process is believed to influence the calibration parameter.

## 5. Final Considerations

The results of two optical fiber cable calibrations are presented. Calibrations of single-mode fiber optic cables were carried out with DSTS (Oz Optics) and DTSS (Sensornet) interrogators. The results of the calibration of tensile tests were performed on fiber optic cables sold by Sensornet and Furukawa. These results were satisfactory and close, demonstrating the possibility of monitoring deformations with a single-mode fiber optic cable. A methodology was also proposed for calibrating fiber optic cable deformation since the calibration parameters provided by the cable manufacturers can result in errors depending on the application. Few fiber optic cables are aimed at civil engineering applications. Therefore, with the calibration methodology proposed herein, we present the possibility of obtaining calibration parameters of any fiber optic cable, even those manufactured for telecommunications purposes and not only for cables for civil engineering use. Geotechnical engineering researchers will thus not need special cables for their applications.

For strain calibration, using the UK cable (Sensornet), the average calibration parameter obtained for the DTSS interrogator is 0.0506 MHz/με; for the DSTS interrogator, it is 0.0511 MHz/με, and the average parameter between the interrogators is 0.0509 MHz/με. The calibration parameter reported by Sensornet is 0.0481 MHz/με. For the Brazilian cable (Furukawa), the average parameter obtained for the DTSS interrogator is 0.0502 MHz/με; for the DSTS interrogator, it is 0.0506 MHz/με, and the average parameter between the interrogators is 0.0504 MHz/με.

With the test arrangement presented, using different data acquisition system technologies, it was possible to obtain simultaneous readings of fiber optic and conventional sensors with compatible results, which allowed for the calibration and verification of the performance of the different types of sensors, optical cables, and data acquisition technologies.

The results show the viability of applying calibrated fiber optic sensors to experimental pile foundations to evaluate the load–displacement behavior of these elements under different loading conditions.

## Figures and Tables

**Figure 1 sensors-25-00324-f001:**
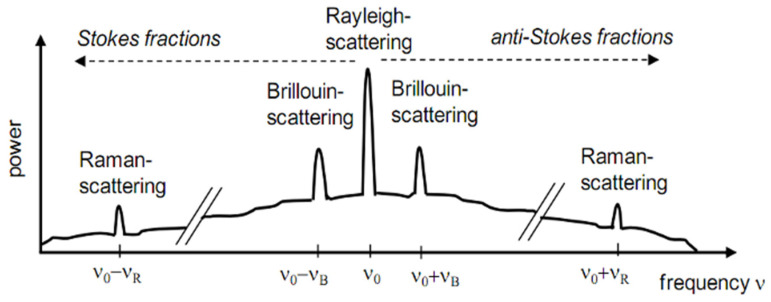
Scattered light spectrum.

**Figure 2 sensors-25-00324-f002:**
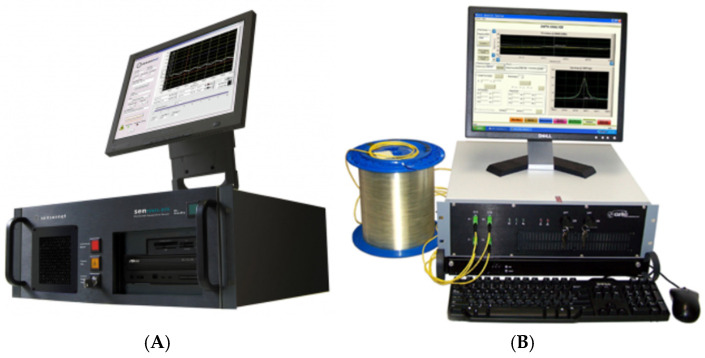
Interrogator: (**A**) DTSS Sensornet and (**B**) DSTS—OZ Optics.

**Figure 3 sensors-25-00324-f003:**
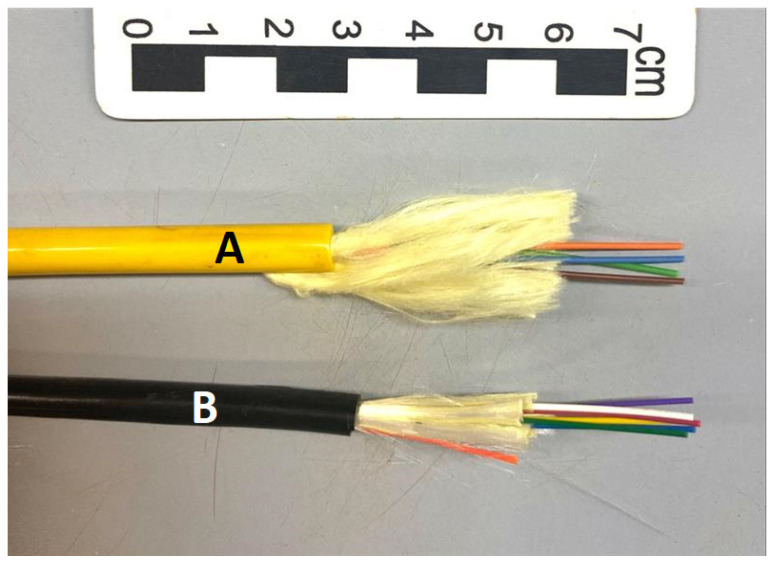
Fiber optic cable: (**A**) Sensornet and (**B**) Furukawa.

**Figure 4 sensors-25-00324-f004:**
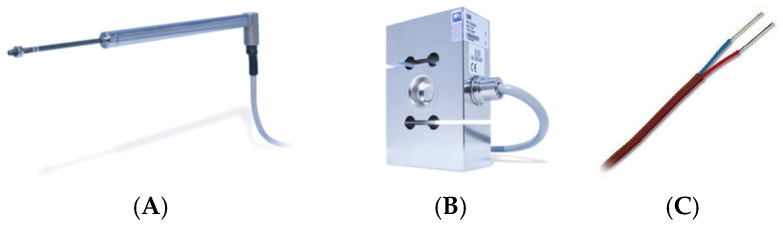
(**A**): Linear displacement transducers; (**B**): HBM load cell; (**C**): T-type thermocouples.

**Figure 5 sensors-25-00324-f005:**
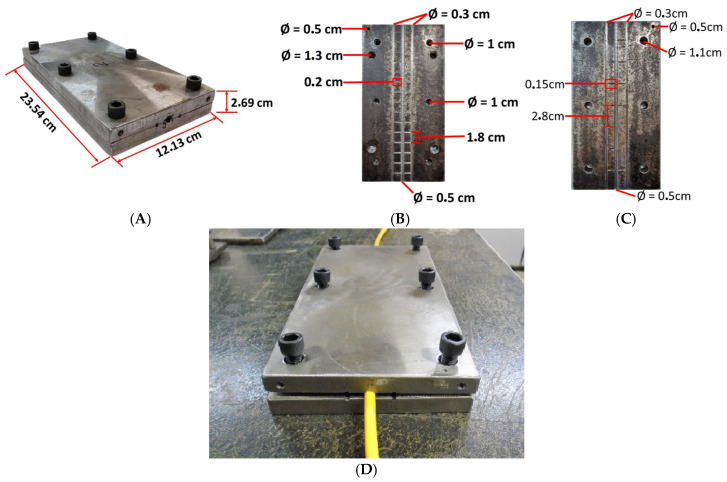
Cable anchoring system. e.g., (**A**) panoramic view, (**B**) superior view, (**C**) inferior view, (**D**) view with the optical cable.

**Figure 6 sensors-25-00324-f006:**
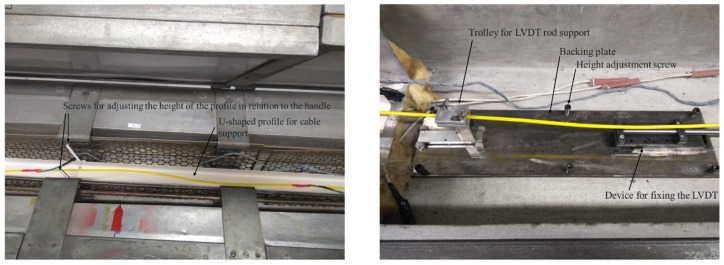
U-shape profile for cable support and the fixation system for LVDTs.

**Figure 7 sensors-25-00324-f007:**
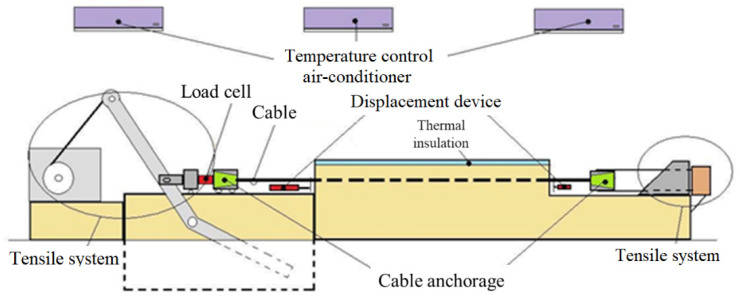
Schematic drawing of the workbench.

**Figure 8 sensors-25-00324-f008:**
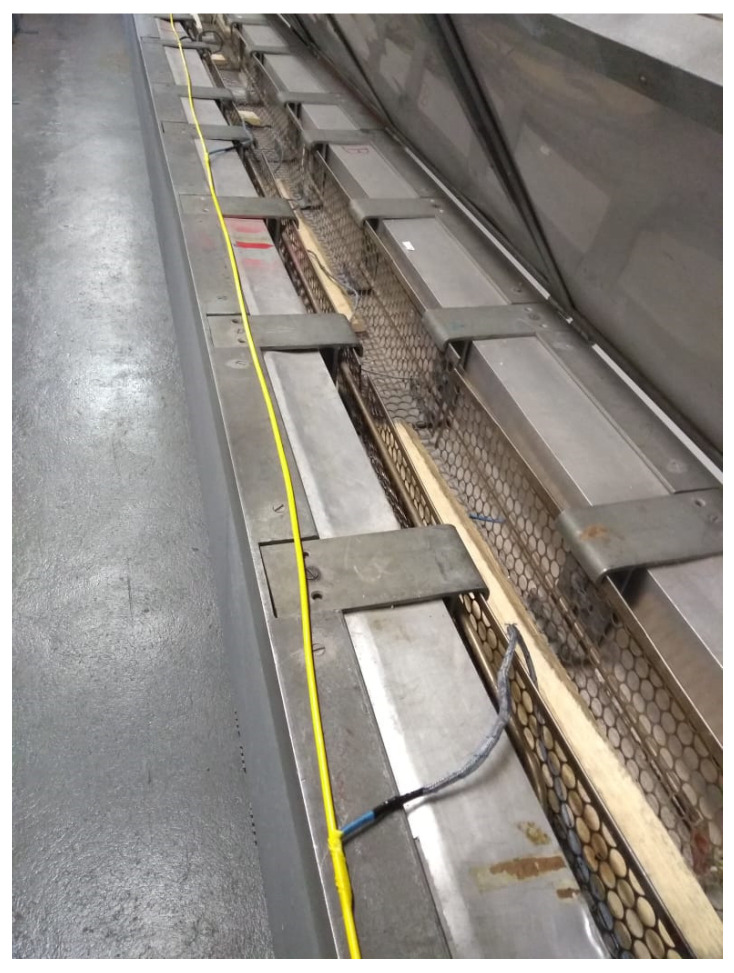
Initial condition of the cable before the beginning of the test.

**Figure 9 sensors-25-00324-f009:**
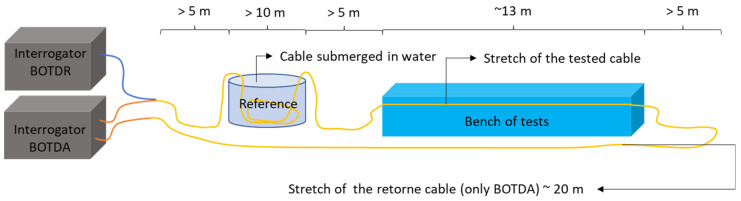
Sketch of the test layout.

**Figure 10 sensors-25-00324-f010:**
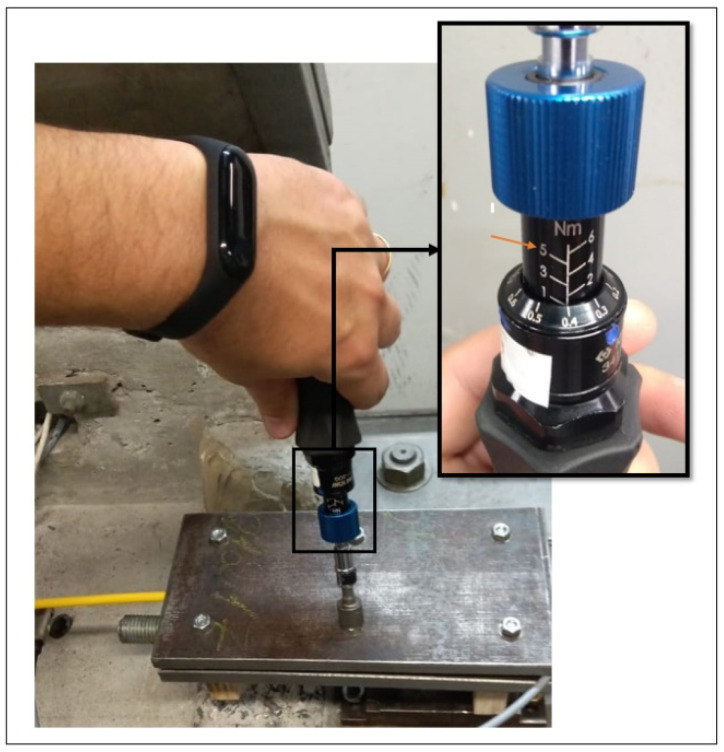
Torquemeter.

**Figure 11 sensors-25-00324-f011:**
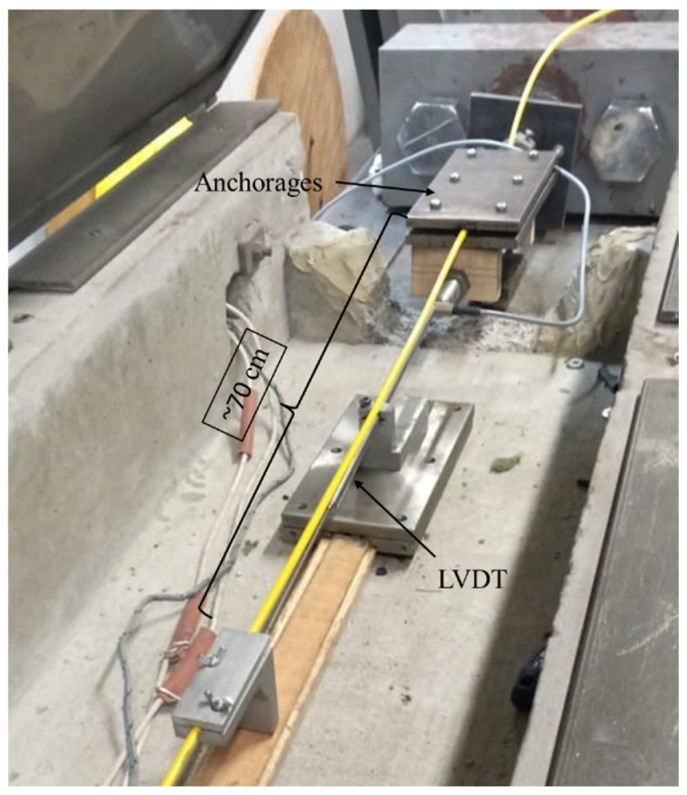
Distance between the displacement sensors and the cable anchorages.

**Figure 12 sensors-25-00324-f012:**
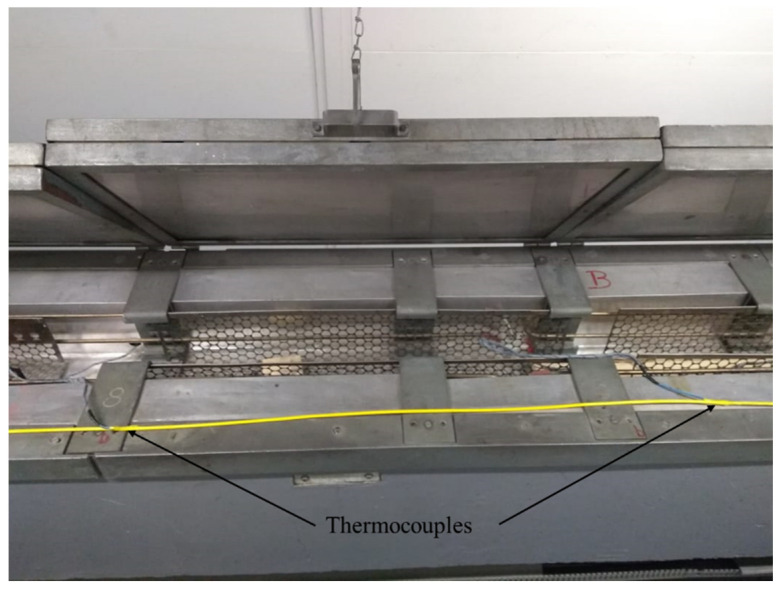
Thermocouples.

**Figure 13 sensors-25-00324-f013:**
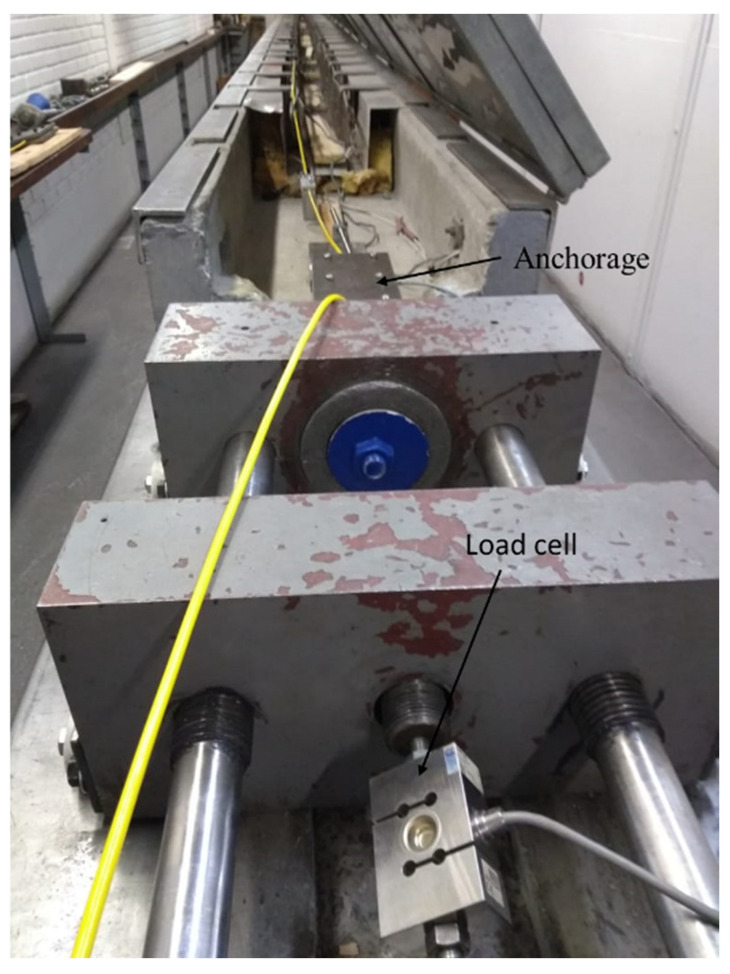
Load cell.

**Figure 14 sensors-25-00324-f014:**
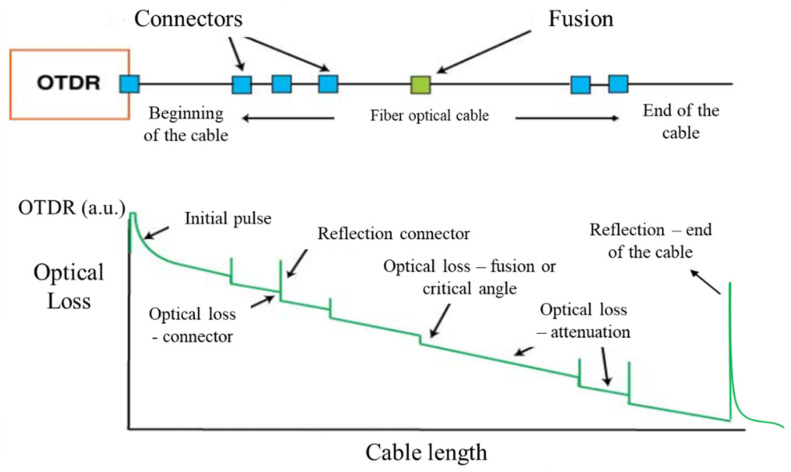
Possible interpretations of the OTDR response signal.

**Figure 15 sensors-25-00324-f015:**
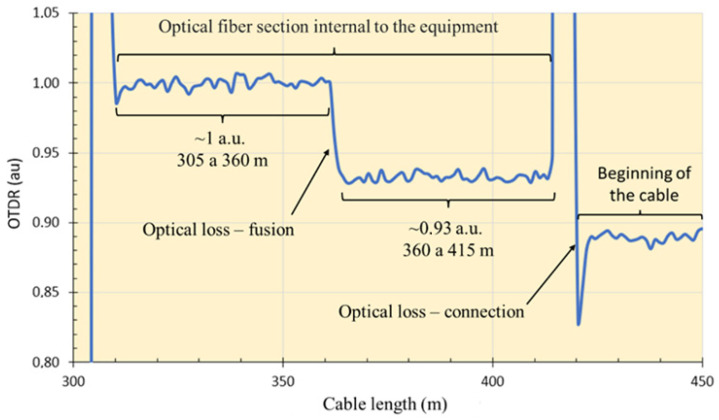
OTDR signal response of the cable stretch inside the interrogator.

**Figure 16 sensors-25-00324-f016:**
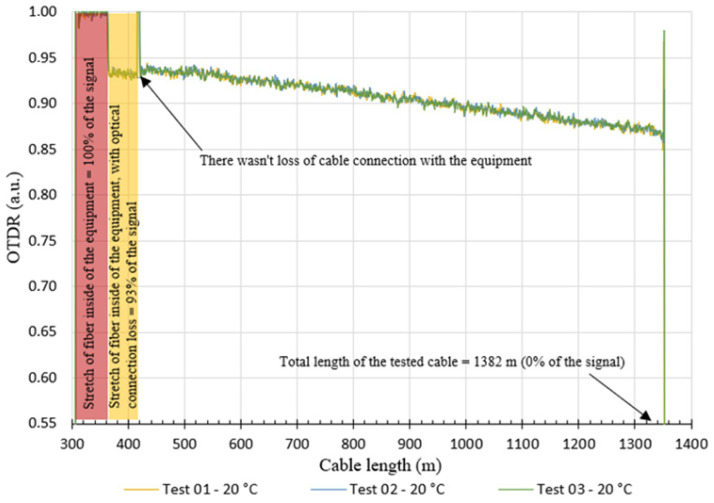
OTDR: Ensaio a 20 °C.

**Figure 17 sensors-25-00324-f017:**
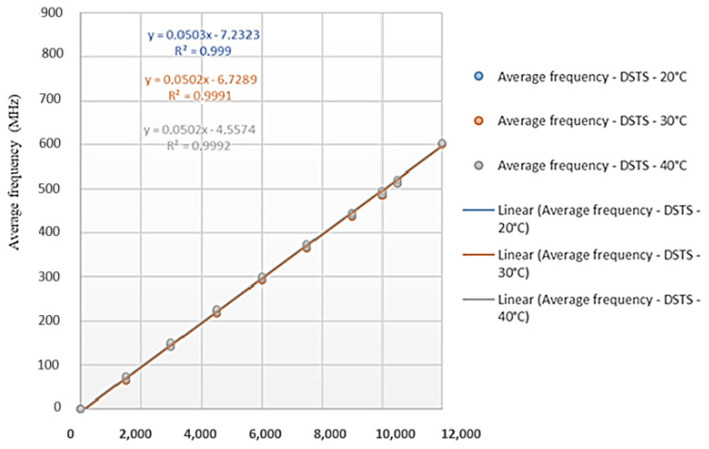
Test results with the DTSS interrogator (Brazilian cable).

**Figure 18 sensors-25-00324-f018:**
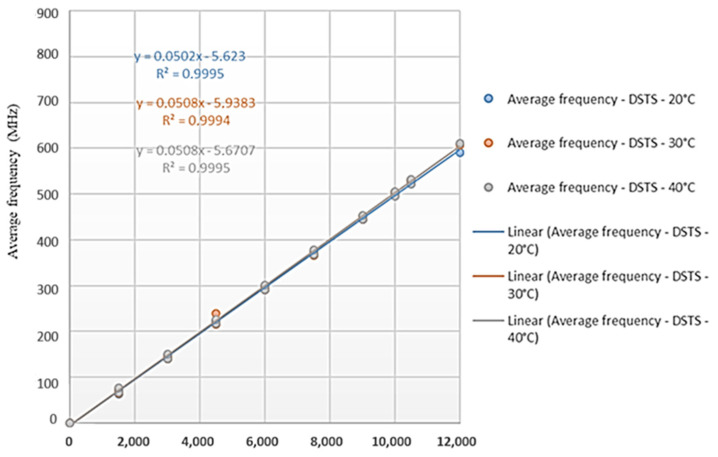
Test results with the DSTS interrogator (Brazilian cable).

**Figure 19 sensors-25-00324-f019:**
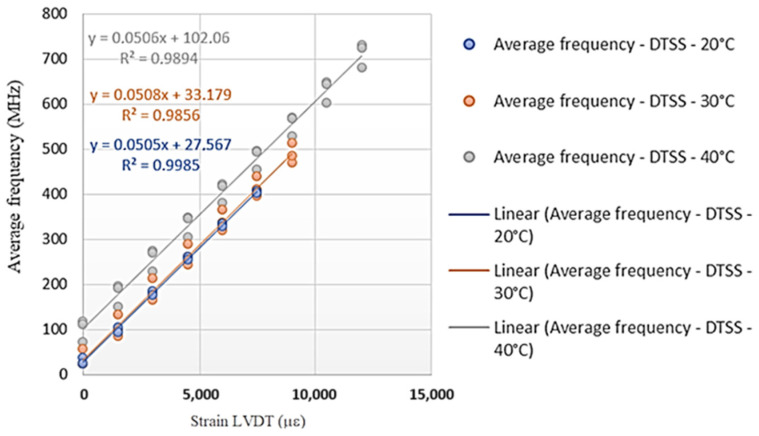
Test results with the DTSS interrogator (Sensornet cable).

**Figure 20 sensors-25-00324-f020:**
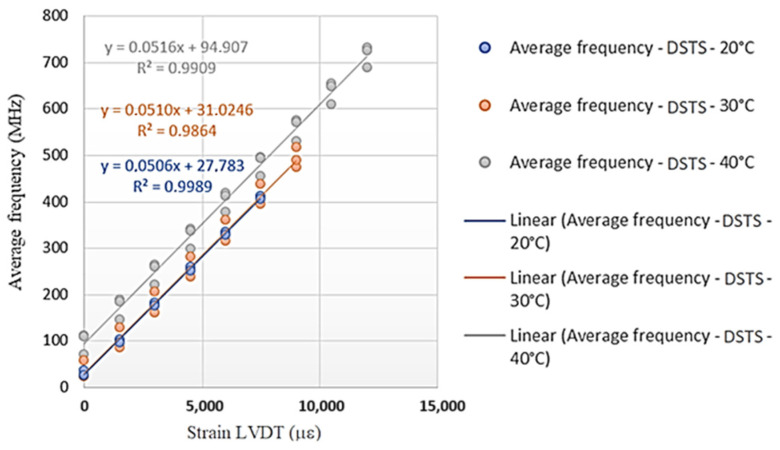
Test results with the DSTS interrogator (Sensornet cable).

**Table 1 sensors-25-00324-t001:** Equipment characteristics.

Parameters		Equipment		
		DTSS	DSTS	
			BOTDR	BOTDA
Resolution of readings	Temperature	1 °C	0.8 °C	0.1 °C
	Strain	10 μξ	16 μξ	2 μξ
Spatial resolution		1.02	1 m–80 m	0.1 m–50 m
Range			0–70 km	0–160 km

**Table 2 sensors-25-00324-t002:** Comparison of test results (Brazilian cable).

Frequency Variation Brillouin [MHz]	Deformation—DTSS [με]			Deformation—DSTS [με]			Difference DTSS [με]	Difference DSTS [με]	Difference DTSS e DSTS [με]
	20 °C	30 °C	40 °C	20 °C	30 °C	40 °C			
	Calibration Data								
	0.0503 MHz/με	0.0502 MHz/με	0.0502 MHz/με	0.0502 MHz/με	0.0508 MHz/με	0.0508 MHz/με			
	Sensornet Calibration Informed								
	0.0481 MHz/με								
0	0	0	0	0	0	0	0	0	0
100	1976	1992	1992	1992	1969	1969	16	23	23
200	3953	3984	3984	3984	3937	3937	31	47	47
300	5929	5976	5976	5976	5906	5906	47	70	70
400	7905	7968	7968	7968	7874	7874	63	94	94
500	9881	9960	9960	9960	9843	9843	79	117	117
600	11,858	11,952	11,952	11,952	11,811	11,811	94	141	141
700	13,834	13,944	13,944	13,944	13,780	13,780	110	164	164
800	15,810	15,936	15,936	15,936	15,748	15,748	126	188	188
900	17,787	17,928	17,928	17,928	17,717	17,717	141	211	211
1000	19,763	19,920	19,920	19,920	19,685	19,685	157	235	235

**Table 3 sensors-25-00324-t003:** Comparison of test results (Sensornet cable).

Frequency Variation Brillouin [MHz]	Deformation—DTSS [με]			Deformation—DSTS [με]			Difference DTSS [με]	Difference DSTS [με]	Difference DTSS e DSTS [με]
	20 °C	30 °C	40 °C	20 °C	30 °C	40 °C			
	0.0506 MHz/με	0.0508 MHz/με	0.0505 MHz/με	0.0516 MHz/με	0.0510 MHz/με	0.0506 MHz/με			
	Sensornet Calibration Informed								
	0.0481 MHz/με								
0	0	0	0	0	0	0	0	0	0
100	1976	1969	1980	1938	1961	1976	12	38	42
200	3953	3937	3960	3876	3922	3953	23	77	84
300	5929	5906	5941	5814	5882	5929	35	115	127
400	7905	7874	7921	7752	7843	7905	47	153	169
500	9881	9843	9901	9690	9804	9881	58	192	211
600	11,858	11,811	11,881	11,628	11,765	11,858	70	230	253
700	13,834	13,780	13,861	13,566	13,725	13,834	82	268	295
800	15,810	15,748	15,842	15,504	15,686	15,810	94	306	338
900	17,787	17,717	17,822	17,442	17,647	17,787	105	345	380
1000	19,763	19,685	19,802	19,380	19,608	19,763	117	383	422

## Data Availability

Data are contained within the article.

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
