# Peer review of "Laboratory Tests Using Distributed Fiber Optical Sensors for Strain Monitoring"

_sensors, 2025, doi:10.3390/s25020324_

Round 1
Reviewer 1 Report (New Reviewer)
Comments and Suggestions for Authors
This work conducted a calibration measurement on optical fibers from two different manufacturers using the Brillouin method. The test is well designed. The materials and method are well illustrated, and the testing results are properly displayed and analyzed.
There are a few points that the authors can be considered:
1) The title of this draft is too broad. There are many more factors that would determine a sensor's on-site application, such as installation, protection, signal transmission and environmental factors. This manuscript only discussed the calibration in laboratory environment. The title is misleading.
2) Introduction section is too long. There is no need to introduce the history of fiber optic sensor. Making a clearer problem statement would be much better for the readers.
3) Section 2 is the principle of Brillouin method. There are other optical fiber strain sensors. FBG fiber uses a different way for strain measurement.
4) The X axis is "Deformation" in Figure 17, 18, 19 and 20, but the unit is micro-strain. Use "Strain" instead of "Deformation" is more accurate.
5) Strongly suggest the authors make the manuscript shorter and more concise by removing some unnecessary details.
Author Response
1) The title of this draft is too broad. There are many more factors that would determine a sensor's on-site application, such as installation, protection, signal transmission and environmental factors. This manuscript only discussed the calibration in laboratory environment. The title is misleading.
New title: Laboratory tests using distributed fiber optical sensors for strain monitoring
2) Introduction section is too long. There is no need to introduce the history of fiber optic sensor. Making a clearer problem statement would be much better for the readers.
Ok, it was reduced.
3) Section 2 is the principle of Brillouin method. There are other optical fiber strain sensors. FBG fiber uses a different way for strain measurement.
It was included: There are other types of measurements, but in this article only what was used to carry out the laboratory tests is discussed.
4) The X axis is "Deformation" in Figure 17, 18, 19 and 20, but the unit is micro-strain. Use "Strain" instead of "Deformation" is more accurate.
Ok, it was changed.
5) Strongly suggest the authors make the manuscript shorter and more concise by removing some unnecessary details.
Ok, it was reduced a little bit.

Reviewer 2 Report (New Reviewer)
Comments and Suggestions for Authors
The submitted article cannot be published in its current form, because it contains excessively redundant information that does not allow to evaluate the scientific content of the article.
1) The paper under review proposes a methodology for calibrating fiber optic cables deformation to show the possibility of obtaining calibration parameters of any fiber optic cable, even those manufactured for telecommunications purposes and not only for cables manufactured for civil engineering use. But the reviewer doesn't see it as novel, because generally speaking, calibration is an important part of strain measurements using fiber optic sensors. The authors didn’t show what a specific gap in the field their study addresses. The authors focused on calibration, but the literature review only cited one papper on calibration ([37]), and that was for strain measurements using Bragg grating fiber optic strain sensors, although the authors did consider strain measurements using distributed fiber optic strain sensing based on Brillouin scattering principle. So, most references are not appropriate. There are several excellent review papers (which are relevant, but not in the reference list of the article under review) on recent distributed optical fiber sensors applications for civil engineering structural health monitoring [https://doi.org/10.3390/s21051818]; on distributed fiber optic sensors for tunnel monitoring [https://doi.org/10.1016/j.jrmge.2024.01.008]; on measurement accuracy, design and calibration [https://doi.org/10.1016/j.sna.2018.11.019]. Perhaps this papers will help the authors formulate a precise answer to the question of what their article adds new compared to other published material and to improve the Introduction.
2) The cited references are not mostly recent publications (within the last 5 years): within the last 5 years — 3 of 40 references.
3) The authors say that 'the results allowed concluding that the application of calibrated fiber optic sensors to experimental piles foundations permits to evaluate the load-displacement behavior of these elements under different loading conditions', but in article named 'A new measurement approach for deflection monitoring of large-scale bored piles using distributed fiber sensing technology' [https://doi.org/10.1016/j.measurement.2017.12.032] it was already shown that measurements obtained from the Brillouin optical time-domain analysis (BOTDA) sensors were found to be in good agreement with the inclinometer data' and 'the BOTDA measurement has great potential to be used for performance monitoring of large diameter piles'.
4) the content of section 2 is unclear for the purposes of the article.
5) section 3 should be drastically shortened.
6) section 4 also needs to be shortened.
Author Response
1) The paper under review proposes a methodology for calibrating fiber optic cables deformation to show the possibility of obtaining calibration parameters of any fiber optic cable, even those manufactured for telecommunications purposes and not only for cables manufactured for civil engineering use. But the reviewer doesn't see it as novel, because generally speaking, calibration is an important part of strain measurements using fiber optic sensors. The authors didn’t show what a specific gap in the field their study addresses. The authors focused on calibration, but the literature review only cited one papper on calibration ([37]),and that was for strain measurements using Bragg grating fiber optic strain sensors, although the authors did consider strain measurements using distributed fiber optic strain sensing based on Brillouin scattering principle. So, most references are not appropriate. There are several excellent review papers (which are relevant, but not in the reference list of the article under review) on recent distributed optical fiber sensors applications for civil engineering structural health monitoring [https://doi.org/10.3390/s21051818]; on distributed fiber optic sensors for tunnel monitoring [https://doi.org/10.1016/j.jrmge.2024.01.008]; on measurement accuracy, design and calibration [https://doi.org/10.1016/j.sna.2018.11.019]. Perhaps this papers will help the authors formulate a precise answer to the question of what their article adds new compared to other published material and to improve the Introduction.
It is not possible specified the gap in the field because each construction has different contour conditions. We will work on filed to have that specification later.
2) The cited references are not mostly recent publications (within the last 5 years): within the last 5 years — 3 of 40 references.
We believe that consideration does not disqualify the article.
3) The authors say that 'the results allowed concluding that the application of calibrated fiber optic sensors to experimental piles foundations permits to evaluate the load-displacement behavior of these elements under different loading conditions', but in article named 'A new measurement approach for deflection monitoring of large-scale bored piles using distributed fiber sensing technology' [https://doi.org/10.1016/j.measurement.2017.12.032] it was already shown that measurements obtained from the Brillouin optical time-domain analysis (BOTDA) sensors were found to be in good agreement with the inclinometer data' and 'the BOTDA measurement has great potential to be used for performance monitoring of large diameter piles'.
4) the content of section 2 is unclear for the purposes of the article.
There are other types measuring principle, but in this article only what was used to carry out the laboratory tests is discussed.
5) section 3 should be drastically shortened.
We think this section should stay like it was developed to allow anyone to reproduce the same work.
6) section 4 also needs to be shortened.
Ok, it was a bit.

Round 2
Reviewer 1 Report (New Reviewer)
Comments and Suggestions for Authors
The revised version has properly addressed the problems and made the manuscript more clear. I have no further comments.
Author Response
Thank you!

Reviewer 2 Report (New Reviewer)
Comments and Suggestions for Authors
The paper under review proposes a methodology for calibrating fiber optic cables deformation to show the possibility of obtaining calibration parameters of any fiber optic cable, even those manufactured for telecommunications purposes and not only for cables manufactured for civil engineering use. The topic is relevant. The content of the article is clearly stated. However, it is necessary to eliminate some shortcomings listed below.
1) How can you explain that in table 2 the values of columns 4, 5 and 6 are the same? And why are the third and fourth columns red, but the sixth column is not? Please write a few words about it.
2) Don't use dot when abbreviating minutes. A unit of measure is "min" or "mins". Please correct in lines 376, 387 and 388.
3) In text and tables, decimals are separated by dots, but decimals on Figures 15–20 are separated by commas. Why? Please use 'dot' in numbers on Figures 15–20.
4) Figures 17−20 are dull and pale. Please use black font, not grey. And change the line colors to more contrasting ones.
Author Response
1) How can you explain that in table 2 the values of columns 4, 5 and 6 are the same? And why are the third and fourth columns red, but the sixth column is not? Please write a few words about it.
The deformation of the Brazilian cable observed using each interrogator was the same at temperature 30°C and 40°C, with means, the temperature at that level doesn’t change the deformation.
This comment was included in the manuscript.
The color was changed to black.
2) Don't use dot when abbreviating minutes. A unit of measure is "min" or "mins". Please correct in lines 376, 387 and 388.
Okay, it was changed
3) In text and tables, decimals are separated by dots, but decimals on Figures 15–20 are separated by commas. Why? Please use 'dot' in numbers on Figures 15–20.
Okay, it was changed
4) Figures 17−20 are dull and pale. Please use black font, not grey. And change the line colors to more contrasting ones.
Okay, it was changed

This manuscript is a resubmission of an earlier submission. The following is a list of the peer review reports and author responses from that submission.
Round 1
Reviewer 1 Report
Comments and Suggestions for Authors
This manuscript reports a test platform for single mode optical cable and the stress test results of two different optical cables by using two distributed sensors. I'm not sure whether the authors think the platform is innovative or the test results are. In my opinion, the presentation is not concise enough, very verbose and unreadable. The author describes too much about the principle, but some knowledge points are not very clear, so that the description is confused. Although the designed platform can realize joint control of temperature and automatic regulation of strain, the difficulty lies in how to realize it in engineering, and there is no breakthrough in principle. Moreover, the discussion of experimental results in this paper is too simple, and the analysis of error sources is all based on speculation, without further experimental verification. On the whole, I think the article is not innovative enough, and the proposed method is not advanced and not easy to reproduce. In addition, the following issues need to be addressed:
1. I think lines 259 to 282 should be placed after line 211, and some of the duplicated content should be eliminated.
2. I think description in lines 212 to 215 are confusing.
3. In Table 1, the table should not span pages, and the decimal point should not use commas (full-text check is recommended for this problem).
4. Figure 8 is mentioned in line 324, but it does not seem to match the content described. Please confirm. In addition, Figure 8 should appear no earlier than Figures 6 and 7.
5. I suggest that the author do not use the two abbreviations DTSS and DSTS, because the sensing technology they refer to is not clear and needs additional explanation. For example, line 380 does not play the function of shortening, so it is better to directly use BOTDR and BOTDA.
6. In line 400, the BOTDA is misspelled.
7. Some picture number described in the article does not match the actual picture number. For example, Figure 19 in line 512 should be Figure 16. I suggest the authors correct similar problems.
Comments on the Quality of English LanguageExtensive editing of English language required
Author Response
Your suggestions were considered and the manuscript is submitted for further review.

Reviewer 2 Report
Comments and Suggestions for Authors
The manuscripts presents the results from an interesting experimental program for evaluating cables strain, but some major points must be reviewed.
All references are not recently (more than 5 years);
The proposed methodology is not clear, should have an item describing its steps;
The suggested use of this cables for monitoring piles foundations seems not to be applicable, since tests were evaluated under tension and piles usually are under compression. Also, for monitoring piles strain must exist a perfect adhesion of the cables to the piles, what is not evaluated in this study.
Considering these points, I suggest a revision of the manuscript, focusing on the methodology of measurement.
Comments on the Quality of English LanguageEnglish is good, minor reviews are necessary.
Author Response

(The authors gave the same response as above.)

Round 2
Reviewer 1 Report
Comments and Suggestions for Authors
Thanks for sending me the revision. I carefully read it, and I am satisfied with the modification based on my opinions.
I think this manuscript can be accepted.
Comments on the Quality of English LanguageThanks for sending me the revision. I carefully read it, and I am satisfied with the modification based on my opinions.
I think this manuscript can be accepted.
Author Response
Thank you for your reviewing.

Reviewer 2 Report
Comments and Suggestions for Authors
All references are not recently (more than 5 years);
Author Response
Thank you for your reviewing.
Three recent references about the use of fiber optics in different applications have been included (17, 25 and 34).
